# Intraocular Sustained Release of EPO-R76E Mitigates Glaucoma Pathogenesis by Activating the NRF2/ARE Pathway

**DOI:** 10.3390/antiox12030556

**Published:** 2023-02-23

**Authors:** Sarah Naguib, Carlisle R. DeJulius, Jon R. Backstrom, Ameer A. Haider, John M. Ang, Andrew M. Boal, David J. Calkins, Craig L. Duvall, Tonia S. Rex

**Affiliations:** 1Neuroscience Program, Vanderbilt University, Nashville, TN 37232, USA; 2Department of Biomedical Engineering, Vanderbilt University, Nashville, TN 37232, USA; 3Vanderbilt Eye Institute, Vanderbilt University Medical Center, Nashville, TN 37232, USA

**Keywords:** NRF2, antioxidant, glaucoma, retinal ganglion cell, neuroprotection, neurodegeneration, oxidative stress, erythropoietin

## Abstract

Erythropoietin (EPO) is neuroprotective in multiple models of neurodegenerative diseases, including glaucoma. EPO-R76E retains the neuroprotective effects of EPO but diminishes the effects on hematocrit. Treatment with EPO-R76E in a glaucoma model increases expression of antioxidant proteins and is neuroprotective. A major pathway that controls the expression of antioxidant proteins is the NRF2/ARE pathway. This pathway is activated endogenously after elevation of intraocular pressure (IOP) and contributes to the slow onset of pathology in glaucoma. In this study, we explored if sustained release of EPO-R76E in the eye would activate the NRF2/ARE pathway and if this pathway was key to its neuroprotective activity. Treatment with PLGA.EPO-E76E prevented increases in retinal superoxide levels in vivo, and caused phosphorylation of NRF2 and upregulation of antioxidants. Further, EPO-R76E activates NRF2 via phosphorylation by the MAPK pathway rather than the PI3K/Akt pathway, used by the endogenous antioxidant response to elevated IOP.

## 1. Introduction

Glaucoma, a leading cause of blindness worldwide, is an age-related progressive optic neuropathy resulting in degeneration and death of retinal ganglion cells (RGCs) [1,2,3]. A major risk factor for glaucoma is sensitivity to intraocular pressure (IOP). Current treatments for glaucoma are directed towards lowering IOP but do not always result in slower progression of the disease. Further, there are issues of poor patient compliance to administering multiple, daily IOP-lowering drops [4,5,6]. Long-term, IOP-independent treatments are needed.

While the exact etiology of glaucoma is still under investivgation by many labs, we know that oxidative stress significantly contributes to disease pathogenesis, and antioxidants have been shown to protect retinal neurons [7,8,9,10,11]. We recently discovered that the retina elicits an antioxidant response to elevated IOP that slows onset of RGC degeneration [11]. This response is regulated by PI3K/Akt mediated phosphorylation of the transcription factor, NRF2, resulting in its translocation into the nucleus and activation of the antioxidant response element (ARE). Previous studies have shown increased levels/activation of NRF2 are neuroprotective in models of optic nerve crush and ischemia/reperfusion injury [12,13].

Erythropoietin (EPO) can decrease oxidative stress through activation of the NRF2/ARE pathway [14,15,16]. EPO is a cytokine that blocks death of red blood cell progenitors. Previous studies have shown that EPO reduces cell death, axon degeneration and oxidative stress in multiple models of glaucoma [17,18,19,20,21]. However, EPO has a half-life of only 7–8 h in vivo [22,23]. To overcome this challenge, our lab used viral delivery of EPO or a mutant form of EPO, EPO-R76E [18,19,21]. EPO-R76E preserves retinal neurons and optic nerve axons, but does not induce a physiologically significant increase in hematocrit [18]. To allow for sustained release but not permanent expression after a single intravitreal injection, we explored microparticle-mediated delivery of EPO-R76E. Poly(lactic-co-glycolic acid)/poly(lactic acid) (PLGA/PLA) microparticles were used to deliver EPO in a rat model of optic nerve crush and demonstrated efficacy in protecting retinal neurons [24]. The authors showed a single injection of these microparticles protected retinal ganglion cells (RGCs) up to 8 weeks after injury, whereas free EPO had to be delivered every 2 weeks to see similar protection. We have also previously protected RGCs using EPO-R76E loaded into PLGA microparticles in a model of blast-induced indirect traumatic optic neuropathy [25]. In that study, we showed that a single intravitreal injection of these microparticles delivered one day after injury protected against optic nerve degeneration and visual function deficits at 4 weeks post injury [25].

In this study, we aimed to determine if EPO-R76E loaded PLGA microparticles (PLGA.EPO-R76E) would protect against downstream neurodegeneration in the well-characterized microbead occlusion model (MOM) of glaucoma. We also sought to elucidate if this neuroprotection occurs via activation of the NRF2/ARE pathway. Our study demonstrates that EPO-R76E activates the NRF2/ARE pathway earlier than the retina’s endogenous activation of NRF2/ARE. Additionally, this activation occurs via a different intermediate signaling molecules than the endogenous antioxidant response to elevated IOP.

## 2. Materials and Methods

PLGA/PLGA.EPO-R76E particles: PLGA particles (10 kDa, 50:50 lactide: glycolide, Sigma) were purchased. EPO-R76E was purified and packaged into PLGA particles as previously described [25].

Mice: C57Bl/6 J (Jackson Labs, Bar Harbor, ME, USA) were group-housed, maintained on a 12-h light-dark cycle, and provided food and water ad libitum. All experiments were approved by the Institutional Animal Care and Use Committee of Vanderbilt University, protocol number M1500029-02. An equal distribution of male and female mice (2–3 months old) were used for this project. Mice were euthanized by anesthetic overdose and cervical dislocation.

Microbead Occlusion: We elevated IOP bilaterally by partial occlusion of the anterior chamber with 2 µL injections of 15-µm diameter FluoSpheres polystyrene microbeads (Thermo Fisher, Waltham, MA, USA) as previously described [11,26]. Additional mice received bilateral injections of an equivalent volume of lactated Ringer’s saline solution as controls. Briefly, 1.5 mm outer diameter/1.12 mm inner diameter filamented capillary tubes (World Precision Instruments, Sarasota, FL, USA) were pulled using a P-97 horizontal puller (Sutter Instrument Company, Novato, CA, USA), and the resulting needles were broken using forceps to an inner diameter of ~100 µm. Microbeads were loaded and injected using a microinjection pump (World Precision Instruments, Sarasota, FL, USA). Mice were anesthetized with isoflurane and pupils were dilated using topical 1% tropicamide ophthalmic solution (Patterson Veterinary, Devens, MA, USA), and 2 µL (~2000 microbeads) were injected. The needle was maintained in the injection site for 20 s before retraction to reduce microbead efflux. Mice were given topical 0.3% tobramycin ophthalmic solution (Patterson Veterinary, Devens, MA, USA) following injection.

Experiment Timeline: One day following bilateral injection of microbeads as described above, mice received intravitreal injection of PBS, empty PLGA particles, or PLGA.EPO-R76E particles. A subset of mice was collected at 2 weeks post-particle injection and another subset was collected at 5 weeks post-particle injection.

IOP measurements: We measured IOP immediately prior to microbead injection and biweekly thereafter using the Icare TonoLab rebound tonometer (Colonial Medical Supply, Franconia, NH, USA) as previously described [21]. Mice were lightly anesthetized using isoflurane, and 10 measurements were acquired from each eye within 4 min of induction of anesthesia. Mice were re-injected with microbeads when the IOPs dropped to naïve levels, usually around 2–3 weeks post-initial IOP elevation, as per previous studies [11].

Construction of AAV2/2.ARE reporter: Synthetic DNA gBlocks were purchased from IDT (Integrated DNA Technologies, Coralville, IA, USA) and contained either wild-type human thioredoxin (Trx) ARE or mutant Trx ARE (oligonucleotide M4) positioned upstream of the mini-TK promoter. The ARE promoters drive expression of nuclear-targeted tdTomato23 and the SV40 promoter drives expression of HA-tagged ZsGreen (Takara Bio, San Jose, CA, USA). pAAV.Trx-ARE(WT)-TnSV0HA-zG was packaged into AAV2/2 (Cell Biolabs, San Diego, CA, USA) and purified by SignaGen (Frederick, MD, USA) resulting in AAV2/2.Trx-ARE(WT)-TnSV0HA-zG (AAV2/2.ARE). A subset of mice were intravitreally injected with AAV2/2.ARE two weeks prior to MOM injections at a concentration of 1 × 10^9^ gc. Imaging was performed 2-weeks post-IOP elevation (4-weeks post AAV2/2.ARE injection).

In vivo electrophysiology: For each experimental condition and at both the 2- and 5-week time points, 6–8 mice were dark-adapted overnight, and the pupils were dilated with 1% tropicamide and anesthetized with 20/8/0.8 mg/kg ketamine/xylaxine/urethane according to the previously published methodology [11,27,28]. Briefly, corneal electrodes with integrated stimulators (Celeris System, Diagnosys LLC, Lowell, MA, USA) were placed on the eyes that were lubricated with GenTeal drops. Subdermal electrodes were placed in the snout and back of the head at the location of the visual cortex. For the VEPs, mice were exposed to 50 flashes of 1 Hz of 0.05 cd.s/m^2^ white light. For the ERGs, mice were exposed to flashes of 1 Hz of 1 cd.s/m^2^ white light. For the photopic negative ERGs (PhNR), mice were exposed to 20 flashes of white light on a green background.

Dihydroethidium (DHE) Fluorescence: DHE was injected intravitreally to detect superoxide, as previously described [11,27]. Six to eight eyes were assessed per experimental condition. Briefly, anesthetized mice were injected with 0.5 µM of DHE (ThermoFisher Scientific, Waltham, MA, USA). For imaging, mice were anesthetized, eyes were dilated, and fluorescence was imaged on a Micron IV retinal imaging microscope (Phoenix Research Labs, Pleasanton, CA, USA). The average intensity of the fluorescence throughout the retina was quantified using ImageJ [29].

MAPK inhibitor experiments: A subset of C57Bl/6J mice were intravitreally injected with a p38 inhibitor, BIRB 796. 1 µL of a 3% drug suspension of 30mg BIRB 796 was intravitreally injected one week after elevation of IOP as previously described [30].

Tissue collection: As previously described, for western blots and qPCR, retinas were collected and flash-frozen. For immunohistochemistry and optic nerve histology, eyes or nerves were collected and post-fixed in 4% paraformaldehyde for 24 h and then transferred to 1X PBS in 4 °C until use.

qPCR: Flash-frozen retinas were homogenized, and RNA was extracted using a Qiagen RNeasy kit (Valencia, CA, USA) according to the manufacturer’s protocol. RNA concentration and purity were measured on a spectrophotometer. First-strand complementary DNA (cDNA) was synthesized using a Superscript III First-Strand synthesis system and oligo-dT20 primers (Invitrogen, Waltham, MA, USA). qPCR was performed using a Power SYBR green master mix (Applied Biosystems, Waltham, MA, USA). All primer sequences were obtained from previous studies (see Table 1) [11]. The assay was performed in triplicate from 4 to 5 separate mice per condition using an Applied Biosciences 7300 real-time PCR system (Waltham, MA, USA). Relative changes in gene expression were determined using GAPDH as the internal control.

Western blot: Protein concentration and Western blot analyses were performed as described previously on the retinas of four to five separate mice per experimental condition [11]. Briefly, a BCA protein assay kit was used to quantify protein levels according to the manufacturer’s protocol (cat#: 23225, ThermoFisher Scientific, Waltham, MA, USA). Flash-frozen retinas were sonicated in a lysis buffer containing 2-ME and heated for 5 min at 95 °C. Known amounts of protein or protein ladder (cat#1610375, Bio-rad, Hercules, CA, USA) were loaded in 4–20% polyacrylamide gels (Bio-Rad #456-1095). After electrophoresis, proteins were transferred onto membranes using a Bio-Rad trans-blot turbo transfer system. Membranes were blocked and then incubated in a primary antibody (see Table 2). After washing, membranes were incubated with an appropriate secondary antibody and then washed and imaged with a Bio-Rad ChemiDoc system. Band density was quantified by scanning the blot using ImageJ in a standard fashion and corrected based on the loading control.

EPO ELISA: EPO-R76E concentration in the cell lysates was measured by sandwich ELISA using a Quantikine Human EPO ELISA kit (R&D Systems, Minneapolis, MN, USA) according to manufacture directions except 25 µL of sample or standard was loaded into each well and 80 µL of EPO Assay Diluent was loaded into each well. Samples were run in duplicate, and results were averaged and interpolated into the standard curve as shown in the kit. To correct for the difference in sensitivity of the EPO ELISA in detecting EPO-R76E, we applied the correction factor, 1.43, calculated in our previous publication [31].

Optic nerve counts: Optic nerves were post-fixed in 4% paraformaldehyde and glutaraldehyde before embedding in Resin 812 and Araldite 502 (cat#: 14900 and 10900 respectively, Electron Microscopy Sciences, Hatfield, PA, USA), as previously described [32]. Briefly, resin-embedded optic nerves were sectioned on a Leica EM-UC7 microtome to a 1 mm thickness. Sections were stained with 1% paraphenylenediamine and 1% toluidine blue and imaged on a Nikon Eclipse Ni-E microscope (Nikon Instruments, Melville, NY, USA). The optic nerves were montaged, and the axons were counted using an ImageJ plug-in. Degenerative axons were identified by dark brown staining due to collapsed myelin or loose myelin (onioning) surrounding the axon. A grid was used to avoid bias by always counting in the same squares, using a cross configuration. Optic nerves from 4 to 5 separate mice per experimental condition were assessed.

Data Analysis: All statistical analyses were performed using GraphPad Prism software (La Jolla, CA, USA). A one-way ANOVA with a Bonferroni post hoc test (a = 0.05) was used to analyze western blot quantification, IHC fluorescence quantification, ON quantification data, and ERG/VEP latencies and amplitudes. A one-way ANOVA and Dunnett’s multiple comparisons post hoc test (a = 0.05) were used to analyze the qPCR results. Means and standard deviation were calculated for each data set.

## 3. Results

### 3.1. PLGA.EPO-R76E Provides Sustained Release of EPO-R76E for up to Six Weeks

After injection of microbeads, IOP remained elevated at a similar level for PBS controls, empty PLGA and PLGA.EPO-R76E groups for the duration of the study (Figure 1A). We injected sufficient PLGA particles to result in 1.65 U/eye of EPO-R76E matching the dose used in our previous studies [25]. This resulted in sustained release of 20–30 mU/mL EPO-R76E in the eye (Figure 1B). Low levels of endogenous EPO were present in eyes injected with empty PLGA particles (Figure 1B).

### 3.2. PLGA.EPO-R76E Is Neuroprotective in the Microbead Occlusion Model of Glaucoma

Elevated IOP causes a decrease in the amplitude of the PhNR of the ERG as early as 2 weeks following microbead injection [11]. The PhNR amplitude remained low in both the PBS controls and the empty PLGA groups at both 2 and 5 weeks post-IOP elevation (Figure 1C). Treatment with PLGA.EPO-R76E prevented this decrease at both 2 weeks (*p* = 0.0093; n = 11 or 14 eyes/group) and 5 weeks post-IOP elevation (*p* = 0.0003; n = 10 or 13 eyes/group; Figure 1C).

Deficits in the amplitudes of the VEP and the b-wave of the ERG (measuring bipolar cell function) have been reported at 4–6 weeks post-IOP elevation [33,34,35]. The ERG b-wave amplitude (b_max_) was similar between PBS controls and PLGA.EPO-R76E injected mice at 2 weeks post-IOP elevation, however mice injected with empty PLGA particles already showed deficits in bipolar cell function (n = 6, 11 or 17 eyes/group respectively; *p* = 0.0097; Figure 1D). At 5 weeks, the mice treated with empty PLGA particles, but not PBS controls or mice treated with PLGA.EPO-R76E, had significantly decreased b_max_ (n = 6, 11 or 17 eyes/group respectively; *p* = 0.0079; Figure 1D). The VEP N1 amplitude was preserved in the PLGA.EPO-R76E treated mice as compared to the empty PLGA particles at both 2 and 5 weeks post-IOP elevation (2 weeks: *p* < 0.0001, n = 8 or 14 eyes/group, respectively; 5 weeks: *p* = 0.0004, n = 10 or 8 eyes/group, respectively; Figure 1E). While there was no difference in the VEP N1 amplitude between PBS controls and PLGA.EPO-R76E treated mice at 2 weeks post-IOP elevation, by the 5 week timepoint, a PBS injection was no longer sufficient to prevent visual function deficits (n = 12 or 8 eyes/group respectively; *p* = 0.0112; Figure 1E).

Degenerative axons were detected in the empty PLGA group at 2 weeks (*p* = 0.0327, n = 5 nerves/group; Figure 1F–H) when axon degeneration is not typically detected in this model [11]. Correspondingly, the total axon count in the empty PLGA injected mice was lower than in saline injected controls in previous studies [11]. At 5 weeks post-IOP elevation, PLGA.EPO-R76E treated mice had few degenerative axons and their total axon counts were in the range of an animal that received no microbead injection (*p* = 0.0027 and *p* = 0.0071, respectively, n = 5 nerves/group; Figure 1G,H). Additionally, at 5 weeks post-IOP elevation, a PBS intravitreal injection results in less axon degeneration than an injection of empty PLGA particles (n = 4 nerves/group; *p* = 0.007; Figure 1H) but more axon degeneration than PLGA.EPO-R76E (n = 4 nerves/group; *p* = 0.0401; Figure 1H).

### 3.3. PLGA.EPO-R76E Reduces Retinal Oxidative Stress In Vivo

Consistent with our previous work, we detected an increase in retinal superoxide following IOP elevation using an DHE as an in vivo marker of hydrogen peroxide and superoxide levels (Figure 2A,B) [11]. At all timepoints examined, PLGA.EPO-R76E treated mice had significantly reduced retinal superoxide levels compared to mice that received empty PLGA particles (Figure 2A,B). The antioxidant effect was the most pronounced at 3 and 4 weeks post-IOP elevation.

### 3.4. PLGA.EPO-R76E Increaseses ARE-Driven Transcripts

We then measured levels of ARE-driven transcripts at 1 and 2 weeks post-IOP elevation using an oxidative stress PCR microarray. Retinas from PLGA.EPO-R76E treated eyes had increases in transcripts from the peroxiredoxin family (Figure 2C), thioredoxin family (Figure 2D), glutathione family (Figure 2E) and oxygen family (Figure 2F) of antioxidant genes. In contrast to our previous studies showing that the retina endogenously upregulates ARE-driven transcripts at 2 weeks post-IOP elevation, treatment with PLGA.EPO-R76E causes increases in ARE-driven transcripts at 1-week post-IOP elevation (Figure 2C–F). We then performed confirmatory qPCR of several of these genes from the microarray. At 1-week post-treatment, treatment with PLGA.EPO-R76E increased transcription of *Txnrd2*, *Ho-1*, *Prdx6*, *Nrf2* and *Sod2* in comparison to injection of empty PLGA particles (*p* = 0.002165, *p* = 0.0259, *p* = 0.00432, *p* = 0.002165, and *p* = 0.002165, respectively; n = 6 retinas/group; Figure 2G). At 2 weeks, we found that only levels of *Txnrd3* and *Ho-1* were upregulated following PLGA.EPO-R76E treatment in comparison to empty PLGA controls (*p* = 0.002165 for both transcripts, n = 6 retinas/group, Figure 2H).

### 3.5. PLGA.EPO-R76E Activates the NRF2/ARE Pathway

We next determined if antioxidant gene expression increases were due to NRF2/ARE activation. The ratio of phosphorylated to total NRF2 was increased as early as 1 week post-IOP elevation in the mice treated with PLGA.EPO-R76E in comparison to empty PLGA controls (*p* = 0.0129; n = 3 retinas/group; Figure 3A,B). To determine activation of the ARE, we intravitreally injected the AAV2/2.ARE reporter two weeks prior to IOP elevation 1 and 2 weeks post-IOP elevation. This reporter fluoresces red when the ARE is activated. AAV2/2, when intravitreally injected, transduces RGCs [6]. At 1 week post-IOP elevation, there was no difference in tdTomato fluorescence in the eyes that received PLGA or PLGA.EPO-R76E (*p* = 0.0545; n = 5 eyes/group; Figure 3C,D). By 2 weeks post-IOP elevation, however, eyes that received PLGA.EPO-R76E had increased tdTomato fluorescence in comparison to eyes that received empty PLGA particles (*p* = 0.0091; n = 5 eyes/group; Figure 3C,D). Using this AAV2/2.ARE reporter, we determine that PLGA.EPO-R76E activates the ARE at 2 weeks post-IOP elevation in RGCs.

### 3.6. PLGA.EPO-R76E Activates NRF2/ARE through the MAPK Pathway

To determine the signaling pathway involved in EPO-mediated phosphorylation of NRF2 in glaucoma, we assessed the ratio of phosphorylated to total PI3K, Akt, JNK, STAT1, STAT3, GSK3β, and MAPK (Figure 4). There was no difference in the ratio of phosphorylated to total PI3K, Akt, JNK (data not shown), STAT1 (data not shown) or STAT3 (data not shown) between eyes that received PLGA or PLGA.EPO-R76E (*p* = 0.7672 for pPI3K/PI3K, *p* = 0.5535 for pAKT/AKT, respectively; n = 3–4 retinas/group; Figure 4A,B). In contrast, there was a decrease in phosphorylated to total GSK3β in the eyes that received PLGA.EPO-R76E (*p* = 0.027; n = 4 retinas/group; Figure 4C). Notably, phosphorylation of GSK3β blocks phosphorylation of NRF2 (Saha et al., 2022). We also measured an increase in phosphorylated to total MAPK in eyes that received PLGA.EPO-R76E (*p* = 0.0485; n = 4 retinas/group; Figure 4D). 

To determine if NRF2 phosphorylation was dependent on MAPK phosphorylation in the presence of PLGA.EPO-R76E, we treated mice with a MAPK inhibitor (BIRB 796) 1 week post-IOP elevation and collected retinas at 2 weeks post-IOP elevation. We showed that repeated injection of vehicle or BIRB 796 did not affect IOP elevation (Figure 5A). We tested if MAPK inhibition would affect visual function. Inhibition of MAPK following PLGA.EPO-R76E treatment decreased PhNR amplitude compared to the vehicle control group (*p* = 0.001; n = 14 and 9 eyes/group respectively; Figure 5B). Vehicle or BIRB 796 treatment did not affect the PhNR amplitude of mice treated with empty PLGA particles (*p* = 0.1552; n = 11 and 9 eyes/group respectively; Figure 5B).

Since we showed PLGA.EPO-R76E’s upregulation of ARE-driven transcripts following IOP elevation, we examined if inhibition of MAPK would affect PLGA.EPO-R76E’s antioxidant properties. Vehicle treatment in PLGA.EPO-R76E treated mice resulted in upregulation of *Txnrd2*, *Txnrd3*, *Prdx6* and *Sod3* (*p* = 0.002165, *p* = 0.00432, *p* = 0.002165, and *p* = 0.002165 respectively; n = 6 retinas/group; Figure 5C). MAPK inhibition prevented this increase in *Txnrd2*, *Ho-1*, *Prdx6*, *Nrf2*, and *Sod2* in PLGA.EPO-R76E injected mice (*p* = 0.0649, *p* = 0.21, *p* = 0.24, *p* = 0.064, *p* = 0.21; n = 6 retinas/group; Figure 5D). This was further confirmed on the protein level with a few representative proteins. BIRB 796 treatment prevented the EPO-R76E driven increase in PRDX6 levels, but did not affect GPX1 or SOD3 (*p* = 0.0532; n = 4 retinas/group; data not shown for GPX1 or SOD3; Figure 5E,F).

To assess NRF2 activation via phosphorylation in the absence of MAPK activity, we examined the ratio of phosphorylated to total NRF2. This ratio was decreased following inhibition of MAPK (*p* = 0.0548, n = 4 retinas/group; Figure 5E,F). These data suggest that PLGA.EPO-R76E’s phosphorylation of NRF2 is dependent on upstream phosphorylation and activation of MAPK.

Lastly, we assessed if inhibition of MAPK would affect axon degeneration at 2 weeks post-IOP elevation. Intravitreal PLGA alone is pro-inflammatory and PLGA.EPO-R76E was sufficient to prevent the decrease in total number of axons (Figure 1F–H) [25]. Vehicle or inhibitor treatment did not affect the total axons or degenerative axons in mice treated with empty PLGA particles (*p* = 0.3796 and *p* = 0.6628 respectively; n = 3 nerves/group; Figure 5G–I). In mice treated with PLGA.EPO-R76E and BIRB 796, there was a decrease in the total number of axons and an increase in the number of degenerative axons in comparison to mice treated with PLGA.EPO-R76E followed by vehicle (*p* = 0.005 and *p* = 0.006, respectively; n = 3 nerves/group; Figure 5G–I).

## 4. Discussion

This study demonstrates that one intravitreal injection of PLGA.EPO-R76E protects against IOP-induced increases in ROS and neurodegeneration for up to six weeks in glaucoma. This is despite evidence from our group and others that PLGA microparticles induce oxidative stress, inflammation, and neurodegeneration [25,36,37]. Our study adds to this body of literature showing that empty PLGA particles increase axon degeneration and decrease visual function. Thus, future studies using other microparticles, perhaps poly(propylene sulfide) particles, to deliver EPO-R76E might be a more therapeutically beneficial option [25].

Here we demonstrate that PLGA.EPO-R76E is neuroprotective and it results in downstream NRF2 phosphorylation, activation of the ARE, and increased antioxidant gene expression at 1- week post-IOP elevation. This is earlier than the retina’s endogenous antioxidant response to ocular hypertension [11]. Others have also shown that EPO can lead to activation of the NRF2 pathway. Genc et al. (2010) showed that EPO added to SH-SY5Y cells resulted in increased *Nrf2* mRNA and NRF2 nuclear translocation. In a cortical lesion model, recombinant EPO treatment upregulated *Nrf2* mRNA as well as *Nqo1*, which is directly regulated by NRF2 [14]. Similar results were found in a rat model of traumatic brain injury [38]. Our study uses an in vivo reporter of ARE activation measured via tdTomato fluorescence. Interestingly, despite the fact that we observe increases in ARE-driven transcripts following PLGA.EPO-R76E treatment at 1 week post-IOP elevation, we do not see increased tdTomato fluorescence until 2 weeks. This could be because our reporter only transduces retinal neurons in the ganglion cell layer, yet, it is possible that PLGA.EPO-R76E is activating the ARE in other cell types at 1 week post-IOP elevation. Previous studies have shown that EPO receptors are found in microglia and astrocytes as well as neurons [39]. Thus, it is feasible that PLGA.EPO-R76E is activating the ARE in glia earlier than what we observe in the GCL neurons that expressed the reporter, but this is outside the scope of this project.

Our study shows that EPO, in the context of glaucoma, induces NRF2 phosphorylation using a different pathway than the retina’s endogenous antioxidant response to elevated IOP. We previously demonstrated that the endogenous antioxidant response of the retina to glaucomatous neurodegeneration is mediated by the PI3K/Akt pathway [11]. In contrast, this study shows that EPO works through the MAPK pathway to activate NRF2. This highlights a role for the MAPK pathway as a potential therapeutic target in glaucoma. This study fits with previous literature, showing that EPO activates the MAPK pathway. In a paper studying hypoxic-ischemic encephalopathy, EPO’s neuroprotective effects were modulated via the phosphorylation of MAPK [40]. In another study using a rabbit model of ischemia-reperfusion injury, EPO’s activation of the Ras/Rac/MAPK pathway was necessary for its protective effects [41]. Using an inhibitor of MAPK, BIRB 796, we determined that inhibition of MAPK prevents EPO’s antioxidant and neuroprotective effects in the microbead occlusion model of glaucoma.

Activation of NRF2 can induce expression of many antioxidants including NADPH oxidases, thioredoxins, heme oxygenase-1 and glutathione peroxidases [42]. We previously showed that elevation of IOP causes an increase in ROS, and phosphorylation of NRF2 resulting in increased expression of multiple Gpx family members, but not Gpx 3, and increased expression of Prdx 4 and 6 [11]. In contrast, activation of the NRF2/ARE pathway by PLGA.EPO-R76E in glaucoma caused increased expression of Gpx 3, Prdx2, and Prdx6. There are likely other differences as well. These results are limited by the transcripts included in the commercial PCR microarray. Thus, treatment with PLGA.EPO-R76E results in activation of an antioxidant profile that is only partially overlapping with the retina’s endogenous antioxidant response to IOP-induced ROS increases.

In conclusion, we demonstrate in this study that PLGA.EPO-R76E administration is a viable option for the protection of the RGCs and vision in a mouse model of glaucoma. We show that PLGA provided sustained release of EPO for up to six weeks from a single intravitreal injection. Additionally, we show that PLGA.EPO-R76E phosphorylates NRF2 via the MAPK pathway. We also showed that NRF2′s phosphorylation and consequential increase in ARE-driven transcripts occurs 1 week earlier than the retina’s endogenous antioxidant response to ocular hypertension.

## Figures and Tables

**Figure 1 antioxidants-12-00556-f001:**
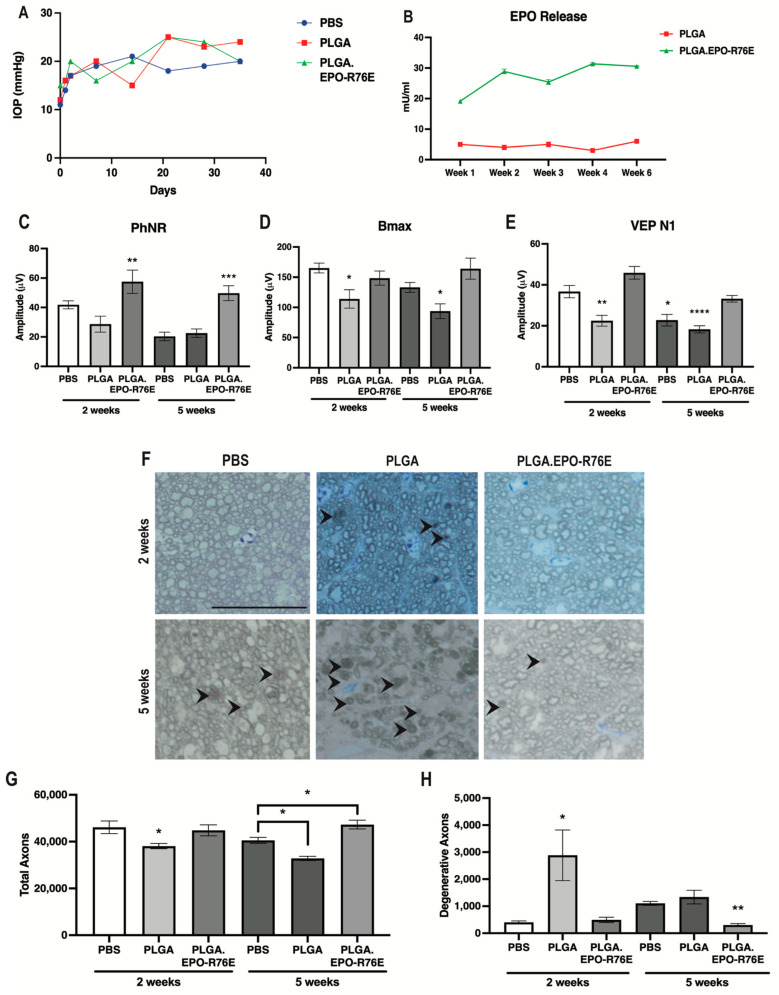
(**A**) IOP levels over time in mice intravitreally injected with PBS, PLGA or PLGA.EPO-R76E. (**B**) Retina EPO levels in PLGA or PLGA.EPO-R76E injected mice at 1, 2, 3, 4, and 6 weeks post-intravitreal injection. (**C**) Quantification of PhNR amplitude of the ERG at 2 and 5 weeks post-IOP elevation for all groups, ** *p* = 0.01, *** *p* < 0.001. (**D**) Quantification of b_max_ amplitude of the ERG at 2 and 5 weeks post-IOP elevation for all groups, * *p* < 0.05. (**E**) Quantification of N1 amplitude of the VEP at 2 and 5 weeks post-IOP elevation for all groups, * *p* < 0.05, ** *p* = 0.01, **** *p* < 0.0001. For all electrophysiological data, comparisons were made between all 2 week groups separately from all 5 week groups; unless otherwise indicated, asterisks indicate differences between PLGA.EPO-R76E group and both PBS and PLGA groups. (**F**) Representative micrographs of optic nerves at 2 and 5 weeks post-IOP elevation for all groups. Line represents 20 μm. (**G**,**H**) Quantification of total and degenerative axons, respectively, in optic nerves at 2 and 5 weeks post-IOP elevation for all groups, * *p* < 0.05, ** *p* = 0.01. For all optic nerve data, comparisons were made between all 2 week groups separately from all 5 week groups; unless otherwise indicated, asterisks indicate differences between PLGA.EPO-R76E group and both PBS and PLGA groups.

**Figure 2 antioxidants-12-00556-f002:**
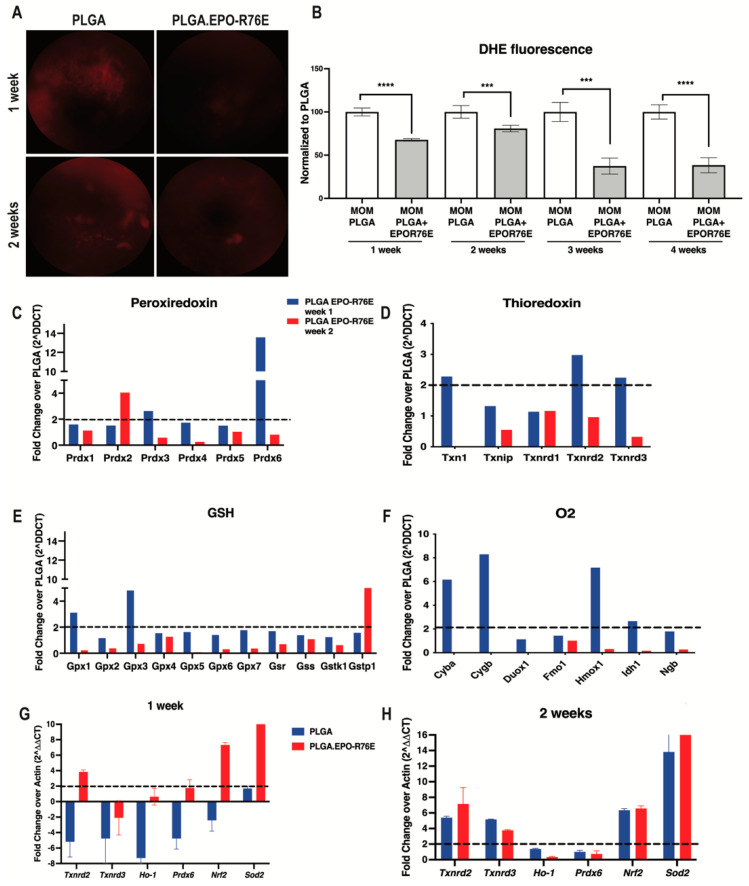
(**A**) Representative fundus images of DHE fluorescence at 1 and 2 weeks post-IOP elevation in empty PLGA controls and PLGA.EPO-R76E injected mice. (**B**) Quantification of DHE fluorescence at 1–4 weeks post-IOP elevation, *** *p* < 0.001, **** *p* < 0.0001. (**C**–**F**) Quantification of peroxiredoxin-related genes, thioredoxin-related genes, glutathione-related genes and oxygen-related genes shown as fold change over PLGA at 1 and 2 weeks post-IOP elevtaion, respectively. (**G**,**H**) Quantification of antioxidant gene transcription shown as fold change over PLGA at 1 and 2 weeks post-IOP elevation, respectively.

**Figure 3 antioxidants-12-00556-f003:**
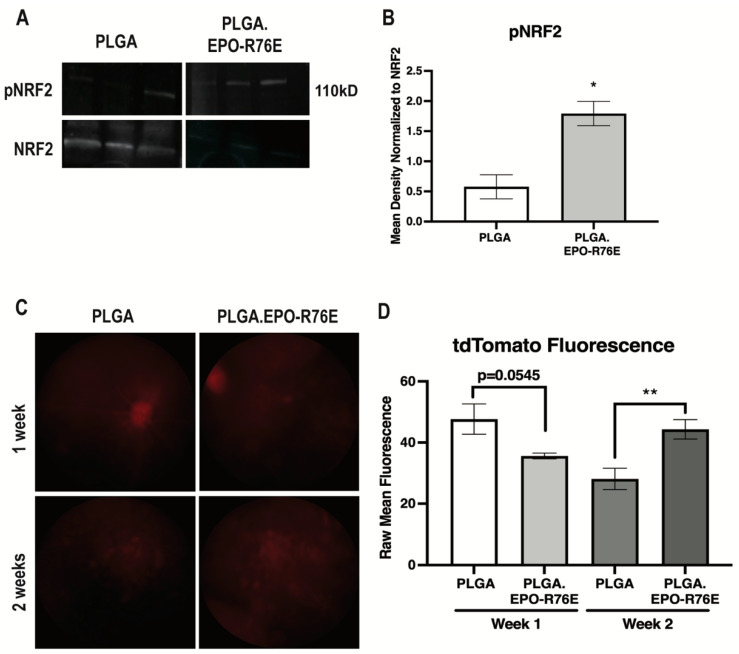
(**A**) Representative Western blot for pNRF2 and NRF2 in PLGA and PLGA.EPO-R76E injected mice at 2 weeks post-IOP elevation. (**B**) Quantification of pNRF2 levels normalized to total NRF2 levels at 1 week post-IOP elevation, * *p* < 0.05. (**C**) Representative fundus images of mice injected with AAV2/2.ARE 2 weeks prior to IOP elevation, imaged at both 1 and 2 weeks post-IOP elevation in mice treated with PLGA or PLGA.EPO-R76E. (**D**) Quantification of tdTomato fluorescence in mice injected with AAV2/2.ARE at 1 and 2 weeks post-IOP elevation, ** *p* = 0.01.

**Figure 4 antioxidants-12-00556-f004:**
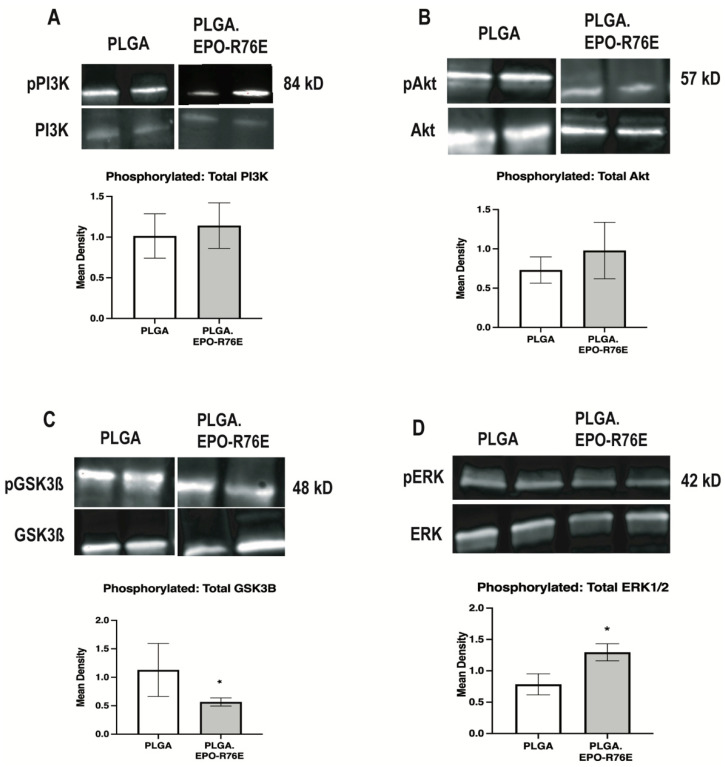
(**A**) Representative Western blot and quantification for pPI3K levels normalized to total PI3K levels. (**B**) Representative Western blot and quantification for pAkt levels normalized to total Akt levels. (**C**) Representative Western blot and quantification for pGSK3β levels normalized to total GSK3β levels. (**D**) Representative Western blot and quantification for pMAPK levels normalized to total MAPK levels, * *p* < 0.05.

**Figure 5 antioxidants-12-00556-f005:**
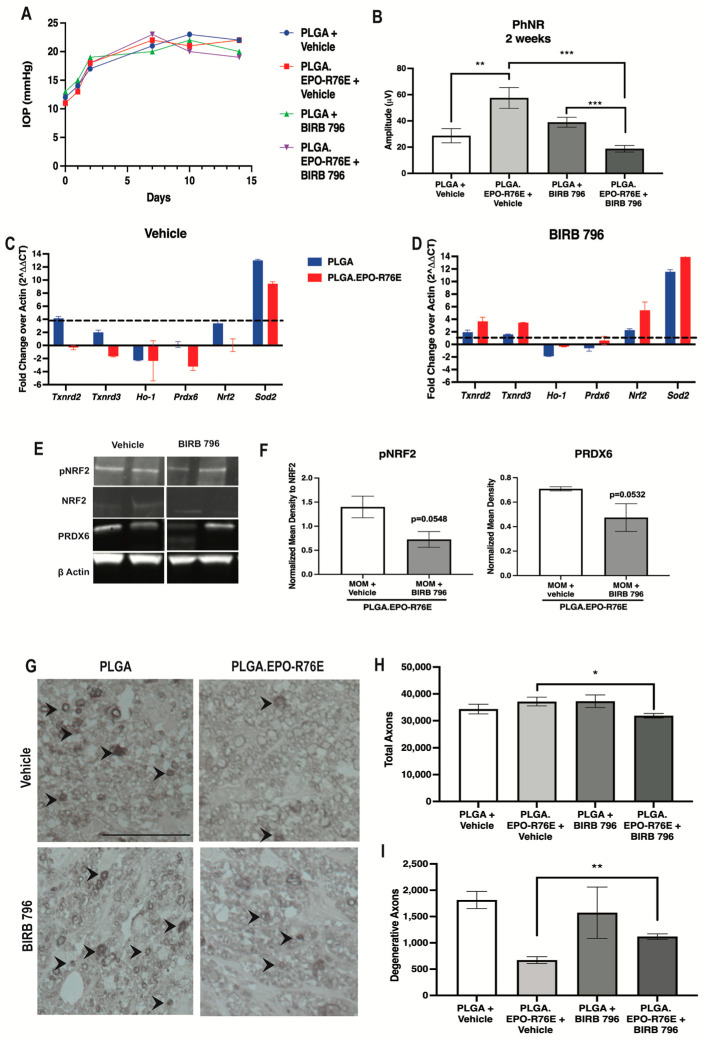
(**A**) IOP levels over time in mice intravitreally injected with PLGA or PLGA.EPO-R76E and then 1 week later, injected with either vehicle or BIRB 796. (**B**) Quantification of PhNR amplitude of the ERG at 2 post-IOP elevation for all groups, ** *p* = 0.01, *** *p* < 0.001. (**C**,**D**) Quantification of antioxidant gene transcription shown as fold change over actin at 2 weeks post-IOP elevation in mice injected with vehicle or BIRB 796, respectively. (**E**) Representative Western blots for pNRF2, NRF2, PRDX6, and β-actin. pNRF2 was normalized to total NRF2 and PRDX6 was normalized to β-actin. (**F**) Quantification of pNRF2 and PRDX6 appropriately normalized as stated in (**E**), * *p* < 0.05, ** *p* = 0.01. (**G**) Representative micrographs of optic nerves at 2 and 5 weeks post-IOP elevation for all groups. (**H**,**I**) Quantification of total and degenerative axons, respectively, in optic nerves at 2 weeks post-IOP elevation for all groups, * *p* < 0.05, ** *p* = 0.01.

**Table 1 antioxidants-12-00556-t001:** qPCR primer sequences.

Gene Name	Forward Primer	Reverse Primer
*Txnrd2*	CGGAGGAACGTGTGTGAATGT	TCAGAGCTTGTCCGAGCAAA
*Txnrd3*	CACGCGGGTTAAGGAACTCTT-	GCCCCGTCATCAACTTGATC
*Ho-1*	CCTTCCCGAACATCGACAGCC	GCAGCTCCTCAAACAGCTCAA
*Prdx6*	TTG ATG ATA AGG GCA GGG AC	CTA CCA TCA CGC TCT CTC CC
*Nrf2*	CCA GCT ACT CCC AGG TTG C	CCA AAC TTG CTC CAT GTC CT
*Sod2*	GAC AAA CCT CAG CCC TAA CG	GAA ACC AAG CCA ACC CCA AC

**Table 2 antioxidants-12-00556-t002:** Table of antibodies used for Western blot analysis.

Protein Name	Catalog Number	Company	Dilution	Species
NRF2	137550	Abcam	1:600	Rabbit
Phosphorylated NRF2 (Ser40 residue)	PA5-67520	ThermoFisher	1:1000	Rabbit
ß actin	4967S	Cell Signaling	1:1000	Rabbit/mouse
PI3K	4257	Cell Signaling	1:1000	Rabbit
Phosphorylated PI3K	4228	Cell Signaling	1:1000	Rabbit
AKT	4685	Cell Signaling	1:500	Rabbit
Phosphorylated AKT	4060	Cell Signaling	1:500	Rabbit
SAPK/JNK	9252	Cell Signaling	1:1000	Rabbit
Phosphorylated JNK	4668	Cell Signaling	1:1000	Rabbit
GSK3ß	12456	Cell Signaling	1:500	Rabbit
Phosphorylated GSK3ß	5558	Cell Signaling	1:2000	Rabbit
MAPK/ERK1/2	4695	Cell Signaling	1:1000	Rabbit
Phosphorylated MAPK/ERK1/2	4370	Cell Signaling	1:1000	Rabbit
Superoxide dismutase 3 (SOD3)	80946	Abcam	1:1000	Mouse
Glutathione peroxidase 1 (GPX1)	PA5-26323	ThermoFisher	1:1000	Rabbit
Peroxiredoxin 6 (Prdx6)	59543	Abcam	1:500	Rabbit

## Data Availability

Original Western blots were provided to the journal prior to review. Any additional data supporting the results shown in this manuscript is available upon request.

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
