# Peer review of "Intraocular Sustained Release of EPO-R76E Mitigates Glaucoma Pathogenesis by Activating the NRF2/ARE Pathway"

_antioxidants, 2023, doi:10.3390/antiox12030556_

Round 1
Reviewer 1 Report
The authors have previously developed PLGA microparticles loaded with a mutant form of erythropoietin (EPO-R76E) and showed the protective activity in animal models of eye disorder. The current study uses EPO-R76E to examine the protective effect of EPO in mice with elevated intraocular pressure. The data are robust and the findings are interesting. The method requires more information. State the animal study ethics number. While the results show the outcome from mice treated with PBS, PLGA and PLGA-EPO-R76E, the treatment schedule is not found in the method. Describe the treatment schedule and state the citation that develops PLGA-EPO-R76E. Is the injection performed prior or after microbead occlusion? What is the termination timepoint? Since the assays are determined at different timepoints, specify the assessment timepoint for each assay in the method. Error bars appear to be missing in Figure 2 C,D,E,F.
Revise Line 41-42 to reflect that the findings are shown in animal models of optic nerve crush and retinal ischaemia and reperfusion.
Lines 91, 114 specify which organ is being dilated.
Author Response
Dear Antioxidants Reviewers,
Thank you so much for taking the time to review our manuscript, and for doing so in such a timely manner. We appreciate your feedback, and we address each of the comments below.
- The animal study ethics number (M1500029-02) has been added to the Methods.
- While the results show the outcome from mice treated with PBS, PLGA and PLGA-EPO-R76E, the treatment schedule is not found in the method. Describe the treatment schedule and state the citation that develops PLGA-EPO-R76E. Is the injection performed prior or after microbead occlusion? What is the termination timepoint? Since the assays are determined at different timepoints, specify the assessment timepoint for each assay in the method.
- We apologize for not including this information in the methods section of the paper. There is a new section now entitled “Experimental Timeline” that address the comments above.
- Error bars appear to be missing in Figure 2 C,D,E,F.
- We apologize for the misunderstanding in this figure. We cannot run statistical analysis in Figures 2C-F, because for the PCR microarray, we had to pool 5 retinas to get enough RNA. Thus, we only ran one technical replicate with 5 samples pooled on each plate, which is why we do not have error bars.
- Revise Line 41-42 to reflect that the findings are shown in animal models of optic nerve crush and retinal ischaemia and reperfusion.
- We had added information in lines 41-42 to emphasize that the findings referred to are in optic nerve crush and ischemia/reperfusion models.
- Lines 91, 114 specify which organ is being dilated.
- We have added clarification that the pupils are being dilated in lines 91 and 114.
Reviewer 2 Report
This is a very interesting study describing the use of EPO-R76E loaded PLGA microparticles for sustained release of EPO-R7E in a microbead occlusion model of glaucoma in rats. The EPO-R76E- PLGA treatment protected RGCs, activated the NRF2/ARE pathway, earlier than the retina’s endogenous activation of this pathway, prevented the increase in retinal superoxide levels, and led to phosphorylation of NRF2 and upregulation of antioxidants. The manuscript is clearly written, and the study was carefully designed and executed.
Please present baseline electrophysiology measurements (prior to glaucoma induction by the injection of the FluoSpheres polystyrene microbeads) of the mice in Figure 1 and 5B. In addition, please present the data of the control mice (injected with lactated Ringer’s saline solution) for the electrophysiology measurement in Figure 1. These controls will enable a better assessment of the therapeutic efficacy of the EPO-R76E loaded PLGA microparticle treatment.
Minor:
Line 49 : please add the actual half-life reported in those studies
Line 141: did you mean C57Bl/6J mice?
Figure 1 panels A and B: I suggest using the same color and shape code for the treatment groups in both panels
Figure 1 all panels: please indicate for each p-value which groups are compared (as you did in panel G for the 5-week time point)
Figure 2 panels C-F : are these results presented as fold change over PLGA (similar to G-H)? Please indicate in the legend text.
Author Response
Dear Antioxidants Reviewers,
Thank you so much for taking the time to review our manuscript, and for doing so in such a timely manner. We appreciate your feedback, and we address each of the comments below.
- Please present baseline electrophysiology measurements (prior to glaucoma induction by the injection of the FluoSpheres polystyrene microbeads) of the mice in Figure 1 and 5B.
- Our goal was to compare between control and treatment groups therefore we did not collect baseline electrophysiology measurements for all mice in Figures 1 and 5B. Previous studies from our lab and others have collected electrophysiological recordings for naïve mice (Alarcón-Martínez et al., 2010; Chrysostomou and Crowston, 2013; Tzekov et al., 2014; Naguib et al., 2020).
- In addition, please present the data of the control mice (injected with lactated Ringer’s saline solution) for the electrophysiology measurement in Figure 1. These controls will enable a better assessment of the therapeutic efficacy of the EPO-R76E loaded PLGA microparticle treatment.
- Since we previously demonstrated that PLGA can have negative effects on the ERG (DeJulius et al., J. Controlled Release, 2021) we chose to compare empty to loaded PLGA particles. This allowed us to investigate the effect of EPO-R76E independent of the effect of the particles and/or injection. We have previously published control electrophysiology measures in Redox Biology (Naguib et al., 2021) at both 2- and 5-weeks post-saline injection.
- Line 49: please add the actual half-life reported in those studies
- We have added the half-life reported in the studies in the introduction as requested.
- Line 141: did you mean C57Bl/6J mice?
- Yes we did, and we have made the change in the methods section to reflect this.
- Figure 1 panels A and B: I suggest using the same color and shape code for the treatment groups in both panels
- We appreciate this comment, and we agree that this would be helpful for readability, so we have made the changes in the figure as suggested.
- Figure 1 all panels: please indicate for each p-value which groups are compared (as you did in panel G for the 5-week time point)
- We apologize for the confusion in this figure—we have changed the information in the legends to reflect the comparisons that are being made.
- Figure 2 panels C-F : are these results presented as fold change over PLGA (similar to G-H)? Please indicate in the legend text.
- We apologize for the confusion in this figure legend. Yes, as in G and H, 2C-F are compared to fold change over empty PLGA particles. We have made the appropriate addition in the text legend.
Round 2
Reviewer 2 Report
The authors addressed all my comments.